# Critical Procedure Identification Method Considering the Key Quality Characteristics of the Product Manufacturing Process

Zhenhua Gao [1], Fuqiang Xu [1,*], Chunliu Zhou [1,2] and Hongliang Zhang [1]

[1] School of Management Science and Engineering, Anhui University of Technology, Ma'anshan 243032, China; seugzh@163.com (Z.G.); clzhou@ahut.edu.cn (C.Z.); hlzhang@ahut.edu.cn (H.Z.)

[2] Key Laboratory of Multidisciplinary Management and Control of Complex Systems of Anhui Higher Education Institutes, Anhui University of Technology, Ma'anshan 243002, China

\* Correspondence: xu_fqahut@163.com

**Abstract:** The product's manufacturing process has an evident influence on product quality. In order to control the quality and identify the critical procedure of the product manufacturing process reasonably and effectively, a method combining genetic back-propagation (BP) neural network algorithm and grey relational analysis is proposed. Firstly, the genetic BP neural network algorithm is used to obtain the key quality characteristics (KQCs) in the product manufacturing process. At the same time, considering the three factors that have an essential impact on the quality of the procedures, the grey correlation analysis method is used to establish the correlation scoring matrix between the procedure and the KQCs to calculate the criticality of each procedure. Finally, taking the manufacturing process of the evaporator as a case, the application process of this method is introduced, and four critical procedures are identified. It provides a reference for the procedure quality control and improvement of enterprise in the future.

**Keywords:** manufacturing process; procedures quality; genetic BP neural network; key quality characteristics; critical procedure

## 1. Introduction

With the change of the VOC (Voice of Customer) environment and enterprise manufacturing mode, the direction of market competition has gradually changed from price to non-price, and product quality plays an essential role in non-price aspects. For manufacturing enterprises, the product formation needs to go through a series of a complex processes such as research, design, manufacturing, use, and feedback. The manufacturing process is one of the most important links. Controlling procedure quality is the core content of quality management in the manufacturing process, and it is also a hotspot of research in the manufacturing field [1,2]. However, the manufacturing process is often composed of many procedures. Monitoring all procedures will require a lot of labor costs and economic costs, and it is difficult to find the main problems.

According to the complexity of the processing procedure in the product manufacturing process, the procedure can be divided into general procedure, special procedure, and critical procedure. Among them, the critical procedure affects the use and function reliability of the product [3]. At the same time, the critical procedure is also the crucial link to manufacturing process quality control, and the effective identification of the critical procedure is the basis of quality control in the product manufacturing process [4]. Therefore, it is necessary to divide the criticality of the procedure. Through the investigation, it is found that most enterprises still have problems in dividing the procedure criticality in China. For example, it mainly relies on product design documents and the experience of product designers, and lacks theoretical support. In this field, relevant scholars have summarized the problem of procedure criticality classification from two aspects [5].

On the one hand, some scholars believe that the identification of critical procedure should be from the perspective of the KQCs of products. The procedure that plays a decisive role in the KQCs of products are determined to be critical procedure [6–8]. There are many QCs (Quality Characteristics) involved in the product formation process, and the importance of these QCs is different. For targeted management, QCs need to be grouped into a certain level. Furthermore, the QCs with the most significant extent are called key QCs [9,10]. Wang et al. proposed an elastic network method to identify KQC in multi-procedure manufacturing, which solved the multiple collinear problems of QCs and was also influential in solving group effects [11]. Jin et al. developed an interval-valued spherical fuzzy ORESTE method based on a 3D mass model to sort KQCs [12]. Ma et al. believed that the identification of KQC helped to reduce the scope of quality detection and improving detection efficiency. They proposed the Mahalanobis–Taguchi System (MTS) based on the RELIEFF algorithm, which combined the least-squares regression with the state-space model to identify KQCs [13]. Wang et al. proposed an improved IGSA algorithm based on reverse learning and immune algorithm, which combined the advantages of filtering efficiency and high-precision packaging to solve the problem of high-quality feature output dimension [14]. The critical procedure determined by this method are subjective and one-sided, ignoring the impact of the manufacturing process on product quality.

On the other hand, some scholars also believe that identifying critical procedure should establish an analysis model of influencing factors between processes, which can calculate the criticality. For this analysis method, some scholars start from different perspectives, such as product design, manufacturing, and quality inspection process [15–17]. Some other scholars have made subsequent studies on this issue. Zheng et al. used the improved quality loss function to focus on identifying the process in the hub assembly procedure, considering cost factors and relative quality loss, laying a foundation for subsequent quality control [18]. Xu et al. proposed the concept of the process node of the ship sub-shop by using the graph theory method and constructed the calculation model of the criticality of the procedure node. One can identify critical procedure by taking correlation degree, quality level, and influence degree as influencing factors among procedure [19]. Latchoumy P et al. calculated the process faults in the grid through the reliability execution model to determine the critical procedure [20]. Yuan et al. used the improved random matrix to establish the association between the manufacturing process and product features. Then, based on the adjusted feature data, the process results were expressed by fuzzy triangular numbers. Finally, one can identify critical procedure based on D-S evidence fusion rules [21]. Although this method considers the factors affecting the procedure, it lacks the analysis of product quality.

Currently, the research on identifying critical procedure in the manufacturing process mainly focuses on these two aspects. However, in the face of changing customer requirements [22], both methods have limitations. As it is subjective and one-sided to identify critical procedure in product manufacturing from only one perspective, product and procedure quality are not considered comprehensively. Therefore, this study proposes a method combining the genetic BP neural network algorithm with grey correlation analysis and comprehensively analyzes the above two angles. Comprehensive analysis of product QCs and product manufacturing process overcomes the problem of singularity and solves the shortcomings in the above problems. At the same time, when obtaining the KQCs of the product, customer requirements are considered to make the results more realistic. In addition, identifying critical procedure is conducive to controlling product quality. The research on quality control mainly focuses on creating quality control charts [23]. Quality data often come from complex processes or uncertain environments, and it is crucial to choose appropriate statistical methods. According to the studies [24–28], neutrosophic statistics can be applied in the industry if fuzziness, uncertainty, and indeterminacy in product quality attribute or control chart parameter or proportion of non-conforming items.

Therefore, it is feasible to use neutrosophic statistics to expand this study, which will be the content of our subsequent research.

The article explains the specific operation process through an enterprise case. According to the final results, the method in the article is feasible, and the results have been affirmed by enterprise managers. Therefore, under the background of product diversification, this study can not only grasp information regarding customer requirements but can also produce products that meet customer requirements based on this information. In addition, identifying critical procedure can better help the manufacturing industry reduce the economic losses caused by the out-of-control process. To reflect the idea of the article, the flow structure of the article is especially drawn, as shown in Figure 1.

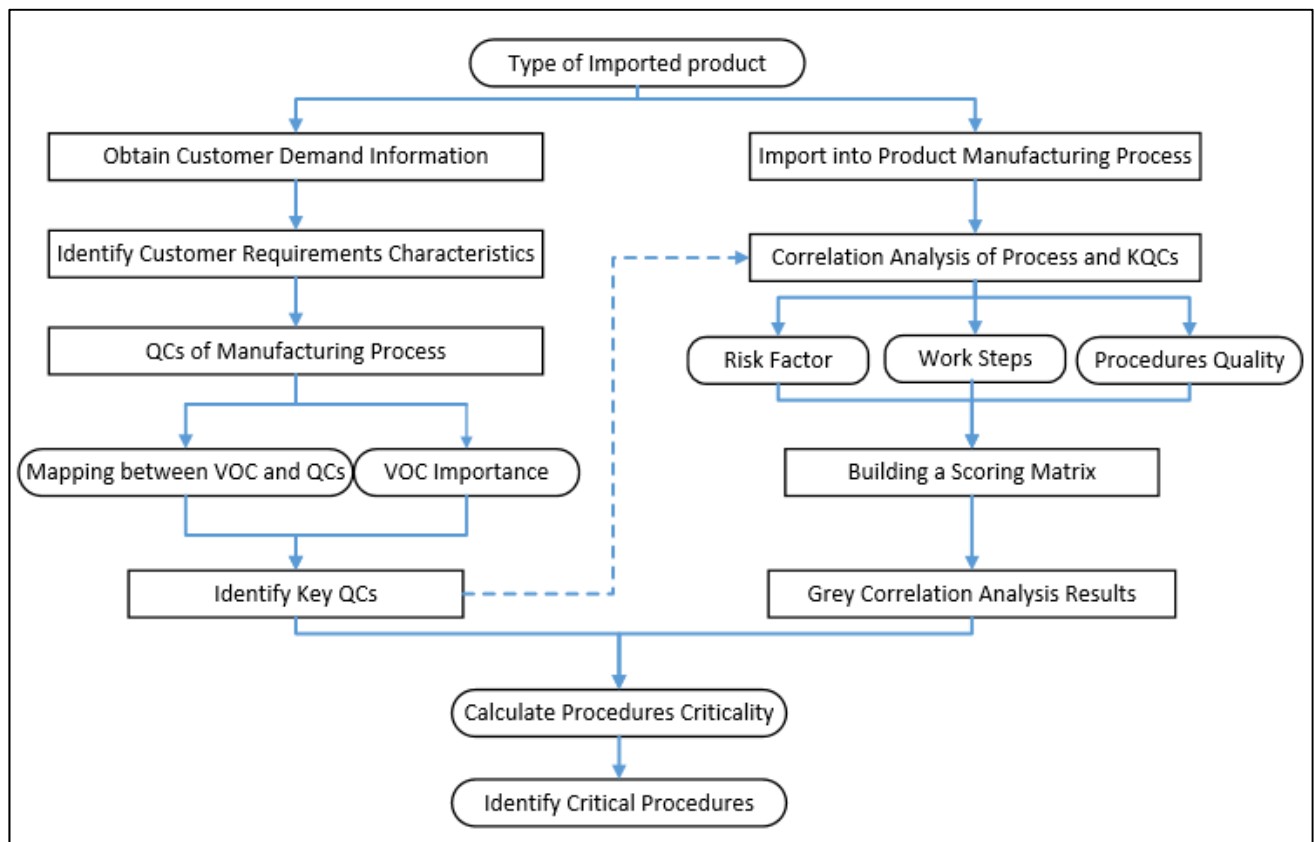

**Figure 1.** Research idea and structure diagram.

## 2. Obtain KQCs with Genetic BP Neural Network

The research object of this section is to obtain the KQCs of products based on customer requirements. Market-owned services and products must meet customers' requirements, which is often mentioned in Six Sigma management. The customer is unprofessional. Only by mapping the requirements reflected by customers to the QCs of the product manufacturing process can we obtain the KQCs of the product more reasonably.

### 2.1. Mapping Process of Product QCs

Using genetic BP neural network algorithm and taking customer requirements as input to obtain the KQCs of products, it is necessary to know the mapping law of QCs in the whole product life cycle, which includes demand analysis, development, and design, production, use, maintenance, feedback, and recovery of products [29].

Product development requires designers to obtain product demand information from customers and their industries and then form customer demand indicators. Through product design, the QCs of customer demand are decomposed, transformed, and reorganized into engineering specifications and requirements for different design stages, forming

the QCs of a design process. The evolution of the QCs of design process to QCs of the manufacturing process is that technicians organize raw material processing, component assembly, and other production tasks according to design documents and information, and finally form the QCs of the manufacturing process. After using the product, customers will provide feedback regarding the experience and opinions to managers, forming customer needs.

As can be seen from Figure 2, product QCs begin with customer demand and end with customer feedback. To obtain the KQCs of the product manufacturing process, you must go through the product design stage: this is a complex many-to-multiple mapping process, that is, between multiple customer demand indicators and multiple quality characteristics, and the problem solving is tedious. To simplify the problems, this section proposes to directly capturing KQCs in the manufacturing process by customer requirements [30].

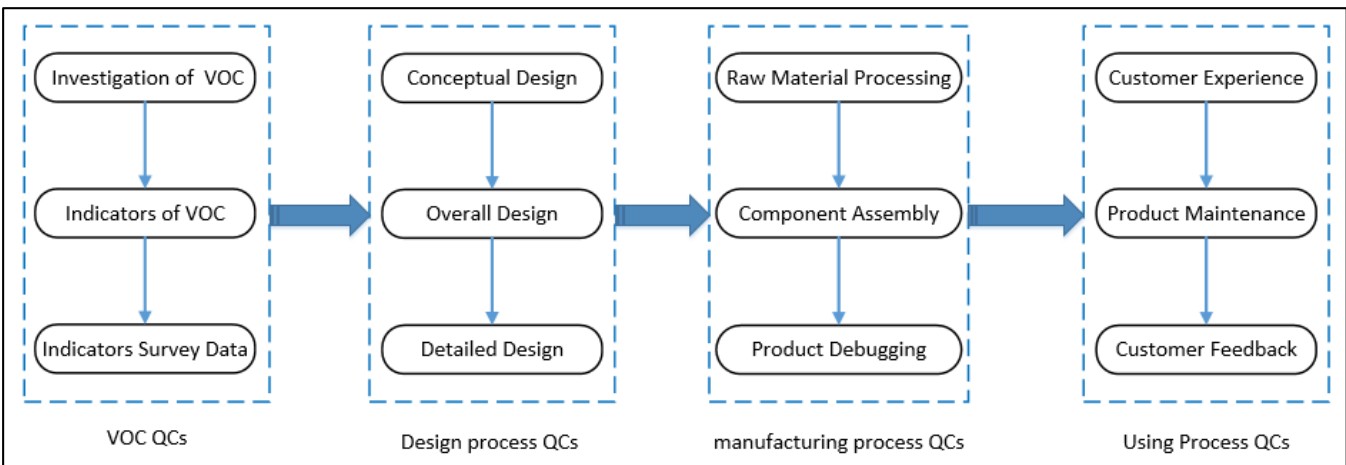

**Figure 2.** Flow chart of the mapping law of quality characteristics.

### 2.2. The Genetic BP Neural Network Theory

In obtaining key quality characteristics in the manufacturing process of products, the process is a complex nonlinear process between multi-input customer requirements and multi-output quality characteristics. For this kind of complex nonlinear mapping problem, it is necessary to use a model that can solve this kind of problem, such as the Kriging model, radial basis function, the BP neural network, etc. However, the Kriging model handles a complicated and large amount of calculation, the radial basis function has high requirements for the selection of central point data, and the parameters are not easy to determine, so it is inconvenient to implement [31–33]. Studies have shown that the error Back-Propagation (BP) algorithm of a 3-layer network structure can approximate any function with arbitrary accuracy to solve complex nonlinear mapping problems [34]. However, BP neural network also has defects, such as poor robustness and sensitivity to different initial connection weights of the network. A genetic algorithm is a kind of algorithm with good global characteristics which can optimize the BP neural network [35]. Therefore, the genetic BP neural network algorithm is more suitable for fitting this relationship.

### 2.2.1. BP Neural Network

The principle of the BP neural network is to reverse update weights and thresholds using the gradient descent method. The mean square error between the expected value and the actual value of the network is minimized. Its 3-layer network structure includes an input layer, an output layer, and a hidden layer, as shown in Figure 3.

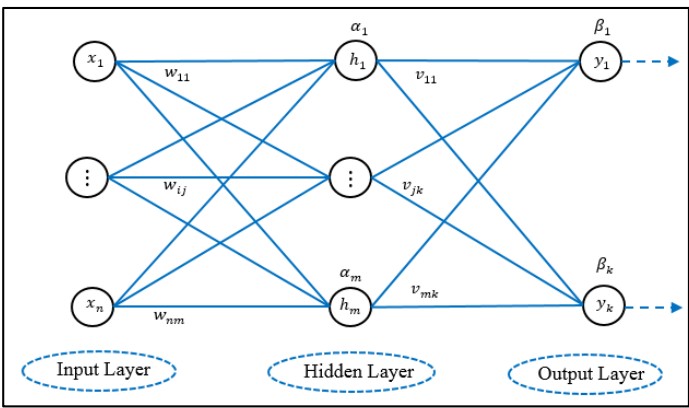

**Figure 3.** Structure diagram of the BP neural network.

In Figure 3, input vector is $X = (x_1, x_2, \cdots, x_n)$, the hidden layer vector is $H = (h_1, h_2, \cdots, h_m)$, the output layer vector is $Y = (y_1, y_2, \cdots, y_k)$, and the desired output vector is $Y' = (y'_1, y'_2, \cdots, y'_k)$. $w_{ij}$ is the connection weight between the input layer and the implied layer. $v_{jk}$ is the connection weight between the implicit layer and the output layer. $\alpha_j$ is the hidden layer threshold, $\beta_k$ is the output layer threshold $(i = 1, 2, \cdots n; j = 1, 2, \cdots, m)$.

2.2.2. Design of Genetic BP Neural Network

- Set up BP neural network: Two main parameters (weight and threshold) of the BP neural network are adjusted by a genetic algorithm. The hidden layer activation function is set as an S-type transfer function, as shown in the equation $f_1(x) = 1/1 + e^{-x}$, the output layer activation function is set to a linear transport function, as shown in the equation $f_2(x) = ax + b$, and take the mean square error as the loss function, as shown in the equation $MSE = \sum_{i=1}^{k} (y_i - y'_i)^2 / k$.

- Initial population: The weights and thresholds of the BP neural network are encoded by actual number coding. The population size is 80 and the evolutionary generation is 100 generations.

- Fitness function of a genetic algorithm: Genetic algorithm takes the individual with the most prominent fitness value as the optimal individual. Therefore, the reciprocal of the mean square error is selected as the fitness function, as shown in Equation (1).

$$Fitness = 1/MSE = k/\sum_{i=1}^{k} (y_i - y'_i)^2 \tag{1}$$

- Genetic operators: Use the most common genetic operators, namely roulette selection, simulated binary crossover, and polynomial mutation operators, using an elite retention strategy. Set crossover probability to 0.8 and mutation probability to 0.1.

*2.3. KQCs in the Product Manufacturing Process*

To obtain the KQCs of the product, the production process is divided into two parts, as shown in Figure 1. Firstly, we should carry out from customer requirements to get the importance of customer requirements indicators, and then get the mapping relationship between customer requirements and quality characteristics. This is based on customer requirements as input and product quality characteristic evaluation value as output. It is further possible to calculate the KQCs of the product manufacturing process.

2.3.1. Calculate Customer Requirements Indicator's Importance

The network model for determining the importance of customer requirements indicators based on the genetic BP neural network is shown in Figure 3. Take customer demand

index as input and the customer total evaluation value as the output. The actual output of the genetic BP neural network model is as follows.

$$output_k = f_2\left[\sum_{k=1}^{q} v_{jk} f_1(\sum_{j=1}^{m} w_{ij} x_i - \alpha_j) - \beta_k\right] \tag{2}$$

MATLAB software trains the network, and the mean square error is calculated. If the mean square error does not meet the set conditions, back propagation is carried out, and the weights and thresholds are updated until the needs are met, and the training stops. The weights and thresholds remain unchanged after the movement. We can use functions to get $w_{ij}$ and $v_{jk}$, and calculate the relative importance of customer demand indicators, as shown in Equation (4).

$$Q_i = \sum_{j=1}^{m} \left| w_{ij} v_{jk} \right| (i = 1, 2, \cdots, n) \tag{3}$$

$$Q_i' = \frac{\sum_{j=1}^{m} \left| w_{ij} v_{jk} \right|}{\sum_{i=1}^{n} \sum_{j=1}^{m} \left| w_{ij} v_{jk} \right|} \tag{4}$$

### 2.3.2. Determination of Mapping Degree

There is a correlation between customer requirements and QCs, and the genetic BP neural network is used to fit the correlation to describe the mapping degree quantitatively. With customer demand index as the re-input of the network model, but with product QCs evaluation value as the output, the mapping degree between customer demand index and QCs is calculated using Equation (5). To objectively obtain the evaluation value of QCs, we compare and score the opinions of most experts and finally get the evaluation value of each quality characteristic in the manufacturing process, using analytical hierarchy process to analyze the opinions of each expert, and the final calculation results are shown in Appendix A.

$$Q_{ik} = \sum_{j=1}^{m} \left| w_{ij} v_{jk} \right| (i = 1, 2, \cdots, n) \tag{5}$$

Calculate the importance of QCs in the production process and determine the KQCs, as shown in Equation (6). Where $A_j$ is the calculation formula of the vital value of KQCs, it is called the relative importance of KQCs. The numerator is the multiplication of the importance of customer requirements and the above mapping. The denominator plays the role of normalization.

$$A_j = \frac{\sum_{i=1}^{n} |Q_i' \times Q_{ik}|}{\sum_{j=1}^{m} \sum_{i=1}^{n} |Q_i' \times Q_{ik}|} (i = 1, 2, \cdots, n; j = 1, 2, \cdots, m) \tag{6}$$

The KQCs of the manufacturing process are determined according to Pareto Principle, which provides the basis for determining the critical procedure in the manufacturing process.

## 3. Identify Critical Procedure in the Manufacturing Process

Product quality and performance are closely related to each procedure in the manufacturing process. Identify the critical procedure in the manufacturing process. The purpose is to control the manufacturing process effectively. On the other hand, as far as possible, reduce the economic and labor cost of quality control.

### 3.1. Determine the Correlation Degree Scoring Matrix

The correlation between KQCs and procedure is ambiguous. According to the basic principle and characteristics of grey correlation analysis, grey correlation analysis is used to evaluate the correlation. Additionally, three influencing factors between processes are considered, combined with the representation of the reachable matrix in the directed graph. Firstly, the reachable matrix of each procedure in the production process is $A = \left[a_{ij}\right]_{n \times n}$. Where $n$ is the number of procedure, the $a_{ij}$ represents the reachable relationship between procedure $i$ and $j$. If the value of $a_{ij}$ is 1, it means reachable between processes; otherwise, it means unreachable.

The impact degree of each procedure in the manufacturing system can be calculated as Equation (7).

$$C_i = \frac{\sum\limits_{j=1}^{n} a_{ij}}{n} \tag{7}$$

In order to facilitate the understanding of the calculation part of the case, it is hereby explained. The procedure reachability corresponding to the KQCs is 1, and the reachability of the other procedure is calculated below. The denominator is the number of KQCs corresponding to the procedure to the beginning and end of the procedure, and the numerator decreases the order of 1 [36]. According to Equation (8), establish the scoring matrix of influencing factors between processes.

$$\text{Correlation Score} = \text{Impact Degree} \times \text{Risk Coefficient} \times \text{Work Steps} \times \text{Procedures Quality} \tag{8}$$

In the above formula, the risk coefficient refers to the risk level information contained in the process related to the manufacturing system. Additionally, it is a measure of process risk in PFMEA. According to the literature [37], its quantitative description is as follows.

$$I_S(v_i) = -\log_2\left[\left(10 - S_{(v_i)}\right)/10\right]$$
$$I_O(v_i) = -\log_2\left[\left(10 - O_{(v_i)}\right)/10\right]$$
$$I_D(v_i) = -\log_2\left[\left(10 - D_{(v_i)}\right)/10\right]$$
$$I(v_i) = I_S(v_i) + I_O(v_i) + I_D(v_i)$$

where S, O, and D respectively represent the severity, probability level, and difficulty of detection of each procedure in PFMEA; $I(v_i)$ is the information content of each procedure. If the procedure contains more information, the higher the risk level. The risk coefficient index is normalized, and the normalized is shown in Formula (9). In this paper, the experimental data of this part are shown in Appendix A.

$$I(v_i)' = \frac{I(v_i)}{10} \tag{9}$$

### 3.2. Grey Correlation Degree Calculation

The principle of grey correlation analysis is to judge whether the sequence curves are closely related according to the similarity of their geometric shapes. The closer the turns are, the greater the correlation between the corresponding sequences. Grey correlation analysis was used to construct the scoring matrix. Complete the following steps.

#### 3.2.1. Determine the Analysis Sequence

The analysis sequence is divided into reference sequence and comparison sequence. Reference sequence can reflect the behavior characteristics of the system and is marked as $Y = (Y_0(1), Y_0(2), \cdots, Y_0(k))^T$; the comparison sequence is a series of factors that affect the behavior of a system, similar to independent variables, and is marked as $X_i = \{X_i(1), X_i(2), \cdots, X_i(k) | k = 1, 2, \cdots, n\}, i = 1, 2, \cdots, m$.

### 3.2.2. Dimensionless Processing of Data

To simplify the calculation, the most commonly used mean change method is adopted in this section.

$$X_i(k)' = \frac{X_i(k)}{\overline{X_i(k)}} \tag{10}$$

where $X_i(k)'$ is the dimensionless value after data processing, $\overline{X_i}(k)$ is the mean value, and $X_i(k)$ is the initial value. Select the maximum value of each row of the association scoring matrix as the optimal value to determine the reference sequence, marked as $Y_0(k) = Max(X_i(k)')$.

### 3.2.3. Calculated Correlation Degree

The results of grey correlation analysis reflect the degree of correlation among different systems under some factors. The absolute difference between each comparison sequence and the reference sequence is calculated. Thus, the minimum and maximum differences at both ends are determined as follows.

$$\min_i \min_k \left| Y_0(k) - X_i(k)' \right|$$
$$\max_i \max_k \left| Y_0(k) - X_i(k)' \right|$$

where $k$ values for $1, 2, \cdots, n$, $i$ values for $1, 2, \cdots, m$. Where $m$ represents the number of comparison sequence objects. According to the Formula (11), the correlation degree between the KQCs and the influencing factors in the procedure is calculated, and the critical procedure identification calculation matrix is determined. Where $\rho$ is the resolution coefficient, evaluated in (0, 1) and normally is 0.5.

$$R_{ij} = \frac{\min\limits_i \min\limits_k \left| Y_0(k) - X_i(k)' \right| + \rho \cdot \max\limits_i \max\limits_k \left| Y_0(k) - X_i(k)' \right|}{\left| Y_0(k) - X_i(k)' \right| + \rho \cdot \max\limits_i \max\limits_k \left| Y_0(k) - X_i(k)' \right|} \tag{11}$$

### 3.3. Calculated Procedures Criticality

The criticality of procedure refers to the importance of procedure in the manufacturing system. After calculating the correlation degree between the procedure and the KQCs, the criticality calculation matrix of the procedures is constructed based on the importance degree of the KQCs, and the calculation results of the criticality are divided according to the Pareto Principle. The criticality calculation matrix is shown in Table 1.

**Table 1.** Calculation matrix of procedures criticality.

| KQCs | KQC 1 | KQC 2 | $\cdots$ | KQC $i$ | Procedures Criticality |
|---|---|---|---|---|---|
| Importance Degree | $w_1$ | $w_2$ | $\cdots$ | $w_i$ | |
| Procedure 1 | $R_{11}$ | $R_{12}$ | $\cdots$ | $R_{1i}$ | $\sum w_i R_{1i}$ |
| Procedure 2 | $R_{21}$ | $R_{22}$ | $\cdots$ | $R_{2i}$ | $\sum w_i R_{2i}$ |
| $\vdots$ | $\vdots$ | $\vdots$ | $\cdots$ | $\vdots$ | |
| $\vdots$ | $\vdots$ | $\vdots$ | $\cdots$ | $\vdots$ | |
| Procedure k | $R_{k1}$ | $R_{k2}$ | $\cdots$ | $R_{ki}$ | $\sum w_i R_{ki}$ |

## 4. Case Study

This section will take the manufacturing process of the evaporator as an example to apply and explain the proposed method. In addition, at the end of this section, the differences between this study and existing studies are discussed. Through analysis, the

results in this paper have been fully affirmed by enterprise managers and proves its effectiveness and applicability. In addition, the customer mainly expounds the customer's demand for evaporator QCs from the three characteristics of practicability, structural elements, and economy, and each feature contains specific demand indicators. At the same time, the manufacturing process of some type of evaporator is given. Figure 4 is a two-dimensional sketch of the evaporator.

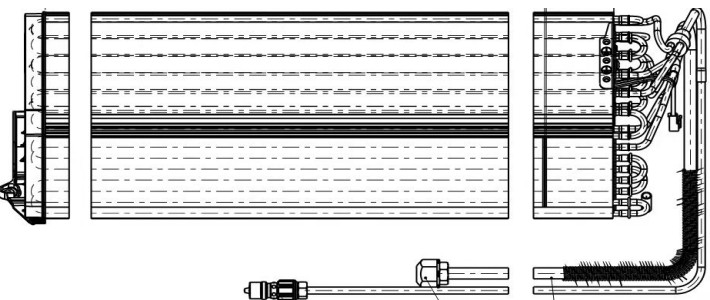

**Figure 4.** Plane diagram of the evaporator.

### 4.1. Obtaining KQCs and Their Importance

Through a survey of customer requirements and enterprise profiles, 40 data samples were obtained. The 40 data samples contain ten customer requirements indicators and 12 QCs, numbered 1–10 and $QC_1$–$QC_{12}$, respectively, as shown in Appendix A.

This study uses three layers of genetic BP neural network structure. The number of input layer neurons is 10, and the number of output layer neurons is 1. However, the number of neurons in the hidden layer cannot be given directly, so we need to determine the value range according to the empirical formula [38,39], which is $m < \sqrt{(n+q)} + a$. Where, $n$ is the number of neurons in input layer, q is the number of neurons in the output layer, and $a$ is the constant with a value range of (1, 10). Therefore, the number of neurons in the hidden layer ranges from $4 < m < 13$. Through the simulation experiment, we determine that when $m = 7$, the genetic BP neural network can achieve the best fitting accuracy. The momentum factor is 0.8, and the learning rate is 0.1. The initial weights and thresholds were optimized by a genetic algorithm with a total size of 80 and 100 iterations. Training samples are used to train the genetic BP neural network. The comparison of training accuracy is shown in Figure 5, and the sample size is 40 groups.

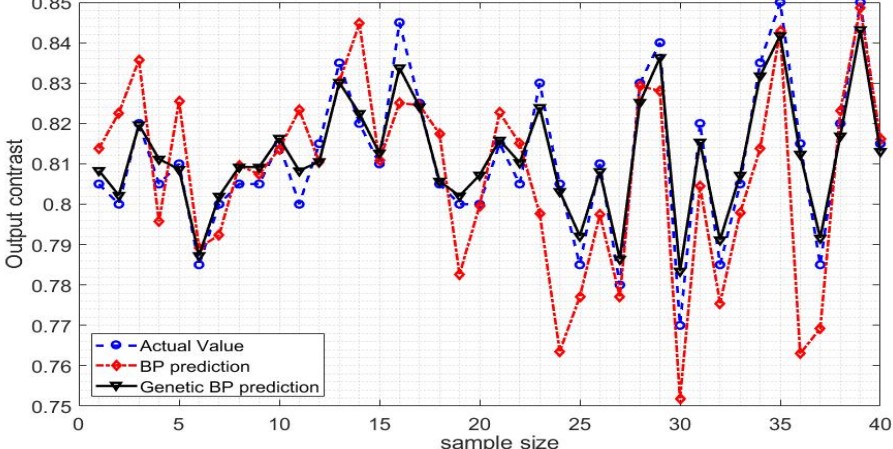

**Figure 5.** Comparison diagram of BP and GABP fitting accuracy.

When the model training is finished, we can get the connection weights of each layer and put the results into Formula (3), from which we can calculate the importance of customer demand indicators. At the same time, it is necessary to build a table that can

reflect the mapping relationship between customer demand indicators and product QCs. As shown in Table 2, the above calculation results of customer demand importance are on the left side of the table. In addition, we also need to know the relationship between customer demand and product QCs. Therefore, it is necessary to change the output in the model and take the evaluation value of product QCs as the output to reconstruct a mapping network that reflect the relationship between customer demand and QCs. However, the connection weights in the network represent the degree of association between customer demand and QCs, which is called the mapping degree. In this way, the degree of mapping between the two relationships is calculated using Formula (5), and the results are filled into Table 2. Finally, based on the above calculation results, the Formula (6) is used to calculate the importance value of product QCs. The last row of the table is the result of the importance calculation of each QC, denoted by $A_j$ in Formula (6). According to the Pareto Principle, we chose the important few as the KQCs. The three KQCs are identified as $QC_3$, $QC_5$, and $QC_8$.

**Table 2.** Table of mapping between VOC and QCs.

| | $Q_i'$ | $QC_1$ | $QC_2$ | $QC_3$ | $QC_4$ | $QC_5$ | $QC_6$ | $QC_7$ | $QC_8$ | $QC_9$ | $QC_{10}$ | $QC_{11}$ | $QC_{12}$ |
|---|---|---|---|---|---|---|---|---|---|---|---|---|---|
| 1 | 0.068 | 0.076 | 0.035 | 0.162 | 0.045 | 0.404 | 0.154 | 0.144 | 0.256 | 0.139 | 0.143 | 0.111 | 0.127 |
| 2 | 0.101 | 0.025 | 0.178 | 0.361 | 0.106 | 0.174 | 0.145 | 0.280 | 0.594 | 0.188 | 0.053 | 0.130 | 0.298 |
| 3 | 0.091 | 0.041 | 0.060 | 0.163 | 0.086 | 0.425 | 0.164 | 0.106 | 0.220 | 0.105 | 0.044 | 0.131 | 0.077 |
| 4 | 0.128 | 0.131 | 0.053 | 0.045 | 0.015 | 0.006 | 0.081 | 0.013 | 0.057 | 0.035 | 0.124 | 0.029 | 0.031 |
| 5 | 0.081 | 0.169 | 0.042 | 0.007 | 0.289 | 0.477 | 0.022 | 0.229 | 0.150 | 0.147 | 0.096 | 0.116 | 0.241 |
| 6 | 0.086 | 0.064 | 0.017 | 0.096 | 0.178 | 0.283 | 0.128 | 0.026 | 0.041 | 0.136 | 0.061 | 0.043 | 0.045 |
| 7 | 0.106 | 0.017 | 0.000 | 0.148 | 0.032 | 0.134 | 0.086 | 0.097 | 0.076 | 0.098 | 0.030 | 0.017 | 0.183 |
| 8 | 0.124 | 0.064 | 0.057 | 0.243 | 0.159 | 0.284 | 0.066 | 0.084 | 0.201 | 0.042 | 0.062 | 0.218 | 0.159 |
| 9 | 0.128 | 0.051 | 0.177 | 0.027 | 0.108 | 0.173 | 0.048 | 0.091 | 0.211 | 0.115 | 0.165 | 0.098 | 0.000 |
| 10 | 0.087 | 0.022 | 0.049 | 0.162 | 0.155 | 0.236 | 0.030 | 0.189 | 0.245 | 0.098 | 0.080 | 0.000 | 0.258 |
| $A_j$ | | 0.045 | 0.049 | 0.096 | 0.077 | 0.165 | 0.061 | 0.082 | 0.138 | 0.072 | 0.060 | 0.062 | 0.092 |

*4.2. Grey Relational Analysis Identifies Key Procedure*

After investigation, it is found that the primary defects of the evaporator include inverted sheet, copper tube, welding, and refrigerant defects. Enterprises mainly rely on subjective experience to control the production process without systematic and scientific identification of critical procedure, resulting in frequent quality problems. Table 3 shows the process flow of the evaporator manufacturing process. In combination with the grey correlation analysis method, this section will analyze the critical degree of each procedure. This will provide a new way for manufacturing enterprises, which can effectively control the production process, especially in the face of a multi-procedure manufacturing process.

**Table 3.** Procedure requirements and data related to the product manufacturing process.

| Serial Number | Name | Cause of Quality | Procedure Quality | Work Step Quantity | Risk Coefficient |
|---|---|---|---|---|---|
| 1 | Aluminum foil fin online | Large-scale rewinding | 21 | 3 | 0.364 |
| 2 | Fixed aluminum foil fin | Copper tube defects | 33 | 2 | 0.306 |
| 3 | Install left bracket | Copper tube defects | 26 | 3 | 0.274 |
| 4 | Fill nitrogen | No nitrogen | 27 | 3 | 0.347 |
| 5 | Copper tube plastic | Copper tube twisted | 24 | 2 | 0.157 |
| 6 | Insert copper tube | Not insert | 18 | 2 | 0.225 |
| 7 | Welding | Welding leakage or blocking | 18 | 4 | 0.306 |
| 8 | Welding inspection | Unchecked | 17 | 2 | 0.177 |
| 9 | Check for fluency | Unchecked | 20 | 3 | 0.321 |
| 10 | Secure hoods | Install the dislocation | 23 | 2 | 0.332 |

| Serial Number | Name | Cause of Quality | Procedure Quality | Work Step Quantity | Risk Coefficient |
|---|---|---|---|---|---|
| 11 | Tie the line | Omit | 16 | 1 | 0.306 |
| 12 | Charge high-pressure test | Unchecked | 19 | 4 | 0.306 |
| 13 | Test it with helium | Unchecked | 26 | 3 | 0.274 |
| 14 | Refrigerant injection | Miss filling refrigerant | 27 | 4 | 0.257 |
| 15 | Install PTC | Large installation error | 12 | 3 | 0.199 |
| 16 | A hot-melt adhesive | Plastic wire drawing | 30 | 4 | 0.364 |
| 17 | Tie the insulation pipe | Omit | 26 | 1 | 0.438 |
| 18 | Install insulation pipe | Not up to requirements | 26 | 2 | 0.284 |
| 19 | Products offline | Damaged | 27 | 2 | 0.232 |

Based on the calculation process of the grey correlation analysis method, firstly, it is necessary to analyze the procedure corresponding to the KQCs. Combined with Formula (7), write the accessibility matrix of each procedure, as shown in Figure 6.

$$
\begin{array}{c}
\text{Procedure} \quad 1 \quad 2 \quad 3 \quad 4 \quad 5 \quad 6 \quad 7 \quad 8 \quad 9 \quad 10 \quad 11 \quad 12 \quad 13 \quad 14 \quad 15 \quad 16 \quad 17 \quad 18 \quad 19 \\
KQC_1 \begin{bmatrix} \frac{1}{14} & \frac{1}{7} & \frac{3}{14} & \frac{2}{7} & \frac{5}{14} & \frac{3}{7} & \frac{1}{2} & \frac{4}{7} & \frac{9}{14} & \frac{5}{7} & \frac{11}{14} & \frac{6}{7} & \frac{13}{14} & 1 & \frac{5}{6} & \frac{2}{3} & \frac{1}{2} & \frac{1}{3} & \frac{1}{6} \\
KQC_2 \quad \frac{1}{7} & \frac{2}{7} & \frac{3}{7} & \frac{4}{7} & \frac{5}{7} & \frac{6}{7} & 1 & \frac{12}{13} & \frac{11}{13} & \frac{10}{13} & \frac{9}{13} & \frac{8}{13} & \frac{7}{13} & \frac{6}{13} & \frac{5}{13} & \frac{4}{13} & \frac{3}{13} & \frac{2}{13} & \frac{1}{13} \\
KQC_3 \quad \frac{1}{9} & \frac{2}{9} & \frac{1}{3} & \frac{4}{9} & \frac{5}{9} & \frac{2}{3} & \frac{7}{9} & \frac{8}{9} & 1 & \frac{10}{11} & \frac{9}{11} & \frac{8}{11} & \frac{7}{11} & \frac{6}{11} & \frac{5}{11} & \frac{4}{11} & \frac{3}{11} & \frac{2}{11} & \frac{1}{11} \end{bmatrix}
\end{array}
$$

**Figure 6.** The reachability matrix of key quality characteristics in the procedures.

The manufacturing process data and the reachability of each procedure have been obtained. Following the order in Section 3.2 above, the grey correlation analysis matrix of procedure and KQCs is determined by Equation (8). The initial scoring matrix can be obtained by dimensionless processing of the data matrix with Equation (10). Finally, the degree of grey correlation is calculated through Formula (11), and the critical procedure is determined accordingly. The final results of the example are shown in Table 4 below. The results show that we have identified four critical procedures, which are step 7, step 9, step 14, and step 16. For enterprises, identifying critical procedure can better control the multifaceted losses caused by product quality problems.

**Table 4.** Calculation result table of procedure criticality.

| | $KQC_1$ | $KQC_2$ | $KQC_3$ | Procedure Criticality |
|---|---|---|---|---|
| **Weighted Value** | **0.0960** | **0.1650** | **0.1380** | |
| 1 | 0.3333 | 0.4228 | 0.4512 | 0.1640 |
| 2 | 0.3437 | 0.4580 | 0.4819 | 0.1751 |
| 3 | 0.3589 | 0.5163 | 0.5310 | 0.1929 |
| 4 | 0.3945 | 0.6971 | 0.6700 | 0.2454 |
| 5 | 0.3421 | 0.4522 | 0.4769 | 0.1733 |
| 6 | 0.3488 | 0.4766 | 0.4978 | 0.1808 |
| 7 | 0.4315 | 1.0000 | 0.8661 | 0.3259 |
| 8 | 0.3486 | 0.4547 | 0.4969 | 0.1771 |
| 9 | 0.4508 | 0.7055 | 1.0000 | 0.2977 |
| 10 | 0.4300 | 0.5719 | 0.7188 | 0.2348 |

**Table 4.** *Cont.*

|  | KQC$_1$ | KQC$_2$ | KQC$_3$ | Procedure Criticality |
|---|---|---|---|---|
| **Weighted Value** | **0.0960** | **0.1650** | **0.1380** | |
| 11 | 0.3522 | 0.4243 | 0.4739 | 0.1692 |
| 12 | 0.5993 | 0.6403 | 0.8541 | 0.2810 |
| 13 | 0.5970 | 0.5663 | 0.7083 | 0.2485 |
| 14 | 0.9097 | 0.5984 | 0.7693 | 0.2922 |
| 15 | 0.3725 | 0.4162 | 0.4620 | 0.1682 |
| 16 | 1.0000 | 0.6153 | 0.8028 | 0.3083 |
| 17 | 0.3697 | 0.4146 | 0.4596 | 0.1673 |
| 18 | 0.3622 | 0.4102 | 0.4533 | 0.1650 |
| 19 | 0.3370 | 0.3948 | 0.4313 | 0.1570 |

### 4.3. Comparison Study

In this section, we will compare the methods in reference [40]. This method belongs to the analysis of problems from the perspective of the manufacturing process. It takes graph theory as the model and determines the critical procedure by constructing the influence degree calculation formula and the unqualified degree calculation formula. With product processing technology as the background, the directional graph model of procedure nodes is drawn, as shown in Figure 7. The above data was used to apply the literature method, and the results were calculated and compared with the results in this paper, as shown in Figure 8.

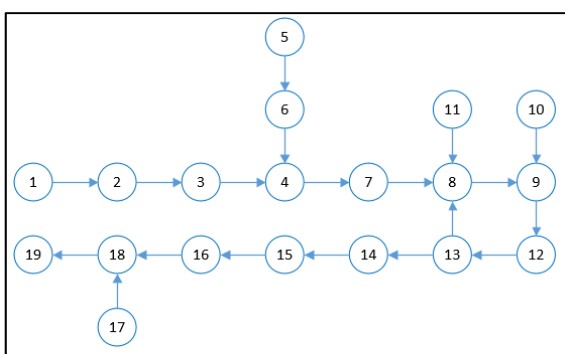

**Figure 7.** Procedures digraph model.

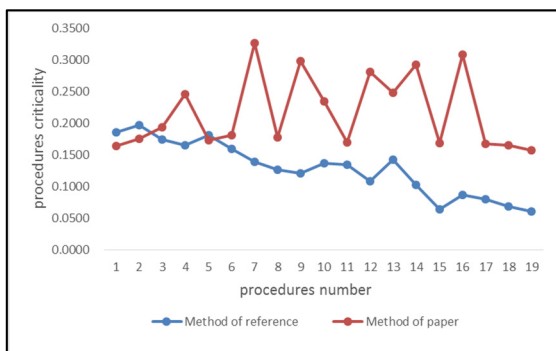

**Figure 8.** A comparison of the two methods.

It can be seen from Figure 8 that the results show a decreasing trend using the methods in the references. This indicates that the starting procedure in the manufacturing process should be identified as the critical procedure. It is not consistent with the actual production situation. However, through comparative research, it is found that the method of this paper takes into account the impact of KQCs on procedure quality and also considers the

influencing factors between processes which make the identification of critical procedure more objective and reasonable.

## 5. Conclusions

In the face of the increasingly fierce market competition, the article takes customer demand as the starting point and proposes a method for quality control of the product manufacturing process. Firstly, the genetic BP neural network is used to fit the correlation degree between customer demand and QCs. Then, obtain the KQCs of the product manufacturing process, and show the specific calculation process and the important calculation results of the KQCs. The results show that the BP neural network optimized by a genetic algorithm can achieve a better fitting effect and improve the accuracy of this calculation. At the same time, the effects of three factors on procedures are considered. The correlation degree matrix is constructed by the grey correlation analysis method to calculate the critical degree of procedure, avoiding the problem of single evaluation factors. Taking the evaporator manufacturing process of an enterprise as a case, the realization process of this method was introduced, and the results indicate the feasibility of the proposed method. In the future, the critical procedure should be paid attention to in the process of product manufacturing to improve product quality effectively.

The purpose of this study is to find out the critical procedure that can affect product quality and function. It can be said that this study is paving the way for the implementation of product quality control methods. However, this study also needs to expand the content of product quality control. At present, the more effective way of quality control is to design a control chart. After reading a large number of literature, it is found that neutrosophic statistics can handle sample data well when it is incomplete, inaccurate, and fuzzy. Therefore, in future research on how to design a quality control chart, we will choose neutrosophic statistics as the research method.

**Author Contributions:** Conceptualization, Z.G., F.X. and C.Z.; methodology, F.X.; software, F.X. and H.Z.; validation, Z.G. and H.Z.; formal analysis, F.X. and C.Z.; investigation, F.X. and H.Z.; resources, Z.G. and C.Z.; data curation, H.Z., F.X. and C.Z.; writing—original draft preparation, Z.G., F.X. and H.Z.; writing—review and editing, F.X. and C.Z.; visualization, Z.G. and H.Z.; supervision, Z.G. and H.Z.; project administration, Z.G., F.X. and H.Z. All authors have read and agreed to the published version of the manuscript.

**Funding:** National Natural Science Foundation of China: [NO. 71772002]; Natural Science Foundation of Anhui Province: [NO. 2008085QG335]; The Open Fund of Key Laboratory of Anhui Higher Education Institutes: [NO. CS2021-03].

**Data Availability Statement:** All the data covered in this article are provided by enterprises or through research, and are displayed in this article.

**Acknowledgments:** The authors would like to thank the professors mentioned in the article for their help in this study, as well as the cases and data provided by the enterprise.

**Conflicts of Interest:** The authors declare no conflict of interest.

## Appendix A

**Table A1.** Product QCs evaluation value.

| Evaluation Value | QC1 | QC2 | QC3 | QC4 | QC5 | QC6 | QC7 | QC8 | QC9 | QC10 | QC11 | QC12 |
|---|---|---|---|---|---|---|---|---|---|---|---|---|
| Sample: 1 | 0.081 | 0.047 | 0.033 | 0.031 | 0.180 | 0.048 | 0.106 | 0.157 | 0.210 | 0.025 | 0.062 | 0.023 |
| 2 | 0.091 | 0.033 | 0.023 | 0.031 | 0.151 | 0.047 | 0.131 | 0.214 | 0.125 | 0.030 | 0.060 | 0.066 |
| 3 | 0.087 | 0.039 | 0.023 | 0.037 | 0.160 | 0.049 | 0.104 | 0.217 | 0.158 | 0.036 | 0.065 | 0.026 |
| 4 | 0.091 | 0.040 | 0.026 | 0.029 | 0.150 | 0.046 | 0.116 | 0.163 | 0.214 | 0.024 | 0.064 | 0.038 |

**Table A1.** *Cont.*

| Evaluation Value | QC1 | QC2 | QC3 | QC4 | QC5 | QC6 | QC7 | QC8 | QC9 | QC10 | QC11 | QC12 |
|---|---|---|---|---|---|---|---|---|---|---|---|---|
| 5 | 0.039 | 0.020 | 0.027 | 0.060 | 0.153 | 0.064 | 0.198 | 0.104 | 0.188 | 0.040 | 0.086 | 0.022 |
| 6 | 0.038 | 0.022 | 0.032 | 0.053 | 0.155 | 0.064 | 0.200 | 0.101 | 0.183 | 0.042 | 0.086 | 0.024 |
| 7 | 0.037 | 0.021 | 0.032 | 0.057 | 0.137 | 0.068 | 0.209 | 0.112 | 0.179 | 0.041 | 0.085 | 0.023 |
| 8 | 0.030 | 0.042 | 0.022 | 0.054 | 0.089 | 0.101 | 0.165 | 0.198 | 0.171 | 0.046 | 0.065 | 0.017 |
| 9 | 0.030 | 0.042 | 0.025 | 0.054 | 0.090 | 0.103 | 0.148 | 0.202 | 0.176 | 0.048 | 0.068 | 0.016 |
| 10 | 0.030 | 0.042 | 0.024 | 0.051 | 0.090 | 0.112 | 0.140 | 0.219 | 0.164 | 0.047 | 0.065 | 0.017 |
| 11 | 0.031 | 0.038 | 0.021 | 0.051 | 0.096 | 0.105 | 0.140 | 0.215 | 0.166 | 0.046 | 0.076 | 0.017 |
| 12 | 0.038 | 0.034 | 0.022 | 0.057 | 0.096 | 0.115 | 0.131 | 0.210 | 0.155 | 0.052 | 0.072 | 0.018 |
| 13 | 0.033 | 0.043 | 0.021 | 0.053 | 0.101 | 0.100 | 0.148 | 0.201 | 0.155 | 0.052 | 0.074 | 0.019 |
| 14 | 0.091 | 0.044 | 0.022 | 0.025 | 0.158 | 0.053 | 0.117 | 0.159 | 0.209 | 0.027 | 0.061 | 0.034 |
| 15 | 0.077 | 0.043 | 0.022 | 0.028 | 0.161 | 0.050 | 0.125 | 0.159 | 0.210 | 0.027 | 0.061 | 0.038 |
| 16 | 0.078 | 0.043 | 0.021 | 0.027 | 0.159 | 0.053 | 0.121 | 0.155 | 0.220 | 0.028 | 0.060 | 0.035 |
| 17 | 0.076 | 0.044 | 0.018 | 0.028 | 0.177 | 0.056 | 0.116 | 0.133 | 0.198 | 0.029 | 0.062 | 0.064 |
| 18 | 0.078 | 0.042 | 0.020 | 0.022 | 0.179 | 0.057 | 0.122 | 0.130 | 0.194 | 0.029 | 0.057 | 0.071 |
| 19 | 0.061 | 0.041 | 0.047 | 0.080 | 0.112 | 0.071 | 0.146 | 0.120 | 0.171 | 0.037 | 0.095 | 0.020 |
| 20 | 0.054 | 0.044 | 0.048 | 0.079 | 0.112 | 0.069 | 0.145 | 0.121 | 0.178 | 0.031 | 0.087 | 0.034 |
| 21 | 0.084 | 0.048 | 0.030 | 0.025 | 0.168 | 0.060 | 0.106 | 0.196 | 0.158 | 0.025 | 0.068 | 0.034 |
| 22 | 0.068 | 0.041 | 0.029 | 0.030 | 0.170 | 0.066 | 0.116 | 0.178 | 0.173 | 0.025 | 0.077 | 0.028 |
| 23 | 0.065 | 0.047 | 0.031 | 0.024 | 0.169 | 0.063 | 0.112 | 0.178 | 0.183 | 0.024 | 0.075 | 0.029 |
| 24 | 0.061 | 0.046 | 0.031 | 0.024 | 0.177 | 0.064 | 0.111 | 0.179 | 0.176 | 0.025 | 0.075 | 0.032 |
| 25 | 0.075 | 0.044 | 0.027 | 0.028 | 0.170 | 0.063 | 0.100 | 0.183 | 0.171 | 0.025 | 0.077 | 0.038 |
| 26 | 0.030 | 0.039 | 0.022 | 0.065 | 0.074 | 0.091 | 0.167 | 0.159 | 0.200 | 0.048 | 0.084 | 0.023 |
| 27 | 0.038 | 0.037 | 0.021 | 0.066 | 0.072 | 0.086 | 0.166 | 0.159 | 0.205 | 0.042 | 0.080 | 0.029 |
| 28 | 0.043 | 0.026 | 0.019 | 0.053 | 0.076 | 0.085 | 0.163 | 0.169 | 0.204 | 0.045 | 0.084 | 0.033 |
| 29 | 0.051 | 0.031 | 0.019 | 0.060 | 0.069 | 0.096 | 0.154 | 0.160 | 0.209 | 0.049 | 0.079 | 0.025 |
| 30 | 0.043 | 0.028 | 0.025 | 0.056 | 0.057 | 0.101 | 0.149 | 0.172 | 0.218 | 0.055 | 0.064 | 0.034 |
| 31 | 0.043 | 0.021 | 0.024 | 0.067 | 0.181 | 0.060 | 0.146 | 0.115 | 0.150 | 0.044 | 0.124 | 0.025 |
| 32 | 0.039 | 0.025 | 0.019 | 0.060 | 0.147 | 0.062 | 0.146 | 0.122 | 0.163 | 0.046 | 0.152 | 0.021 |
| 33 | 0.038 | 0.019 | 0.021 | 0.059 | 0.165 | 0.063 | 0.133 | 0.141 | 0.156 | 0.045 | 0.135 | 0.026 |
| 34 | 0.039 | 0.023 | 0.019 | 0.060 | 0.155 | 0.066 | 0.142 | 0.149 | 0.153 | 0.043 | 0.117 | 0.035 |
| 35 | 0.073 | 0.024 | 0.020 | 0.051 | 0.132 | 0.070 | 0.131 | 0.158 | 0.156 | 0.034 | 0.118 | 0.033 |
| 36 | 0.066 | 0.040 | 0.022 | 0.024 | 0.157 | 0.050 | 0.118 | 0.202 | 0.182 | 0.031 | 0.074 | 0.036 |
| 37 | 0.061 | 0.039 | 0.020 | 0.024 | 0.149 | 0.051 | 0.120 | 0.211 | 0.182 | 0.031 | 0.079 | 0.034 |
| 38 | 0.063 | 0.042 | 0.020 | 0.023 | 0.153 | 0.063 | 0.113 | 0.190 | 0.184 | 0.035 | 0.083 | 0.031 |
| 39 | 0.060 | 0.038 | 0.026 | 0.026 | 0.147 | 0.064 | 0.110 | 0.215 | 0.180 | 0.029 | 0.074 | 0.033 |
| 40 | 0.064 | 0.036 | 0.032 | 0.035 | 0.148 | 0.064 | 0.111 | 0.198 | 0.174 | 0.031 | 0.075 | 0.033 |

**Table A2.** Values of risk coefficient correlation calculation parameters.

| Procedure Number | 1 | 2 | 3 | 4 | 5 | 6 | 7 | 8 | 9 | 10 | 11 | 12 | 13 | 14 | 15 | 16 | 17 | 18 | 19 |
|---|---|---|---|---|---|---|---|---|---|---|---|---|---|---|---|---|---|---|---|
| S | 6 | 6 | 4 | 4 | 3 | 3 | 5 | 3 | 4 | 5 | 3 | 4 | 4 | 4 | 6 | 4 | 6 | 5 | 5 |
| O | 5 | 5 | 5 | 7 | 4 | 4 | 4 | 2 | 4 | 5 | 3 | 5 | 5 | 3 | 5 | 5 | 6 | 3 | 2 |
| D | 6 | 4 | 5 | 5 | 2 | 5 | 6 | 4 | 7 | 6 | 4 | 6 | 5 | 4 | 6 | 3 | 7 | 6 | 5 |

**Table A3.** Customer demand index survey data.

| VOC Indicators Survey | $VOC_1$ | $VOC_2$ | $VOC_3$ | $VOC_4$ | $VOC_5$ | $VOC_6$ | $VOC_7$ | $VOC_8$ | $VOC_9$ | $VOC_{10}$ |
|---|---|---|---|---|---|---|---|---|---|---|
| Sample: 1 | 0.85 | 1.00 | 0.90 | 0.85 | 0.90 | 0.75 | 0.60 | 0.70 | 0.85 | 0.65 |
| 2 | 1.00 | 1.00 | 0.85 | 0.80 | 0.75 | 0.70 | 0.65 | 0.75 | 0.75 | 0.75 |
| 3 | 0.90 | 0.95 | 0.95 | 0.90 | 0.95 | 0.65 | 0.75 | 0.80 | 0.70 | 0.65 |

**Table A3.** *Cont.*

| VOC Indicators Survey | VOC$_1$ | VOC$_2$ | VOC$_3$ | VOC$_4$ | VOC$_5$ | VOC$_6$ | VOC$_7$ | VOC$_8$ | VOC$_9$ | VOC$_{10}$ |
|---|---|---|---|---|---|---|---|---|---|---|
| 4 | 0.85 | 0.95 | 0.90 | 0.85 | 0.85 | 0.65 | 0.70 | 0.85 | 0.85 | 0.60 |
| 5 | 0.80 | 0.95 | 0.90 | 0.85 | 0.80 | 0.70 | 0.60 | 0.85 | 0.80 | 0.85 |
| 6 | 0.85 | 0.90 | 0.90 | 0.80 | 0.85 | 0.70 | 0.55 | 0.75 | 0.75 | 0.80 |
| 7 | 0.80 | 0.85 | 0.85 | 0.80 | 0.80 | 0.75 | 0.75 | 0.85 | 0.85 | 0.70 |
| 8 | 0.75 | 1.00 | 1.00 | 0.75 | 0.95 | 0.65 | 0.60 | 0.90 | 0.70 | 0.75 |
| 9 | 0.90 | 1.00 | 0.95 | 0.80 | 0.90 | 0.65 | 0.65 | 0.85 | 0.70 | 0.65 |
| 10 | 0.85 | 1.00 | 0.85 | 0.85 | 1.00 | 0.70 | 0.65 | 0.90 | 0.75 | 0.60 |
| 11 | 0.85 | 1.00 | 0.85 | 0.90 | 0.95 | 0.70 | 0.70 | 0.75 | 0.75 | 0.55 |
| 12 | 1.00 | 0.90 | 0.90 | 1.00 | 0.85 | 0.75 | 0.65 | 0.70 | 0.80 | 0.60 |
| 13 | 0.80 | 0.95 | 0.90 | 0.85 | 0.90 | 0.60 | 0.70 | 1.00 | 0.85 | 0.80 |
| 14 | 0.75 | 0.95 | 0.95 | 0.95 | 0.75 | 0.75 | 0.70 | 0.85 | 0.80 | 0.75 |
| 15 | 0.85 | 0.95 | 1.00 | 0.90 | 0.85 | 0.80 | 0.65 | 0.75 | 0.75 | 0.60 |
| 16 | 0.90 | 0.90 | 0.85 | 0.85 | 0.85 | 0.85 | 0.75 | 0.80 | 1.00 | 0.70 |
| 17 | 0.95 | 1.00 | 0.85 | 0.85 | 0.80 | 0.75 | 0.70 | 0.85 | 0.85 | 0.65 |
| 18 | 0.90 | 0.95 | 0.80 | 0.80 | 0.80 | 0.70 | 0.70 | 0.90 | 0.75 | 0.75 |
| 19 | 0.80 | 0.90 | 0.95 | 0.95 | 0.85 | 0.70 | 0.55 | 0.85 | 0.80 | 0.65 |
| 20 | 0.80 | 0.95 | 0.95 | 0.90 | 0.90 | 0.65 | 0.65 | 0.80 | 0.80 | 0.60 |
| 21 | 0.85 | 0.95 | 0.90 | 0.90 | 0.95 | 0.65 | 0.70 | 0.85 | 0.75 | 0.65 |
| 22 | 0.75 | 1.00 | 0.85 | 0.80 | 1.00 | 0.60 | 0.65 | 0.95 | 0.75 | 0.70 |
| 23 | 1.00 | 1.00 | 0.90 | 0.80 | 0.85 | 0.75 | 0.60 | 0.90 | 0.85 | 0.65 |
| 24 | 0.90 | 0.85 | 0.95 | 1.00 | 0.80 | 0.70 | 0.55 | 0.85 | 0.85 | 0.60 |
| 25 | 0.85 | 0.95 | 0.85 | 0.75 | 0.85 | 0.75 | 0.60 | 0.80 | 0.80 | 0.65 |
| 26 | 0.90 | 0.95 | 1.00 | 0.85 | 0.85 | 0.65 | 0.55 | 0.75 | 0.85 | 0.75 |
| 27 | 0.90 | 0.90 | 0.95 | 0.80 | 0.90 | 0.60 | 0.60 | 0.70 | 0.75 | 0.70 |
| 28 | 0.90 | 0.90 | 1.00 | 0.95 | 0.95 | 0.65 | 0.75 | 0.85 | 0.70 | 0.65 |
| 29 | 0.85 | 0.95 | 0.90 | 1.00 | 0.90 | 0.60 | 0.70 | 0.85 | 1.00 | 0.65 |
| 30 | 0.75 | 0.85 | 0.85 | 0.80 | 0.90 | 0.65 | 0.65 | 0.80 | 0.85 | 0.60 |
| 31 | 0.85 | 0.90 | 0.95 | 0.85 | 0.85 | 0.70 | 0.65 | 0.90 | 0.80 | 0.75 |
| 32 | 0.80 | 0.90 | 1.00 | 0.75 | 0.85 | 0.75 | 0.60 | 0.75 | 0.75 | 0.70 |
| 33 | 0.85 | 0.95 | 0.85 | 0.70 | 0.80 | 0.70 | 0.70 | 0.95 | 0.80 | 0.75 |
| 34 | 0.85 | 0.95 | 0.90 | 0.90 | 0.90 | 0.65 | 0.70 | 1.00 | 0.85 | 0.65 |
| 35 | 0.90 | 1.00 | 0.90 | 0.85 | 1.00 | 0.70 | 0.75 | 0.85 | 0.90 | 0.65 |
| 36 | 1.00 | 0.95 | 0.85 | 0.80 | 0.85 | 0.65 | 0.60 | 1.00 | 0.85 | 0.60 |
| 37 | 0.95 | 0.90 | 0.85 | 0.85 | 0.80 | 0.75 | 0.65 | 0.80 | 0.75 | 0.55 |
| 38 | 0.80 | 0.95 | 0.90 | 1.00 | 0.85 | 0.80 | 0.55 | 0.80 | 0.85 | 0.70 |
| 39 | 0.85 | 1.00 | 1.00 | 0.95 | 0.90 | 0.75 | 0.70 | 0.90 | 0.80 | 0.65 |
| 40 | 0.85 | 0.95 | 0.95 | 0.80 | 0.85 | 0.75 | 0.65 | 0.85 | 0.75 | 0.75 |
| Output: VOC comprehensive evaluation | 0.800 | 0.815 | 0.835 | 0.820 | 0.810 | 0.845 | 0.825 | 0.805 | 0.800 | 0.800 |

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
