# Peer review of "Critical Procedure Identification Method Considering the Key Quality Characteristics of the Product Manufacturing Process"

_processes, doi:10.3390/pr10071343_

Round 1

Reviewer 1 Report

After reading through the manuscript, I found myself really difficult to understand this manuscript. The English writing is too horrible for a proper reviewing process. 

Author Response

We revised the manuscript according to the comments of reviewers. Please see the appendix for details.

Reviewer 2 Report

The authors have conducted a comprehensive study of s a method combining a genetic BP neural network with grey correlation analysis. The different processing technologies were discussed. The BP neural network optimized by the genetic algorithm can achieve a better fitting effect. The purpose and necessity of this study are fully defined. This paper is in good shape and can be accepted. However, in Fig. 5, what’s the unit for sample size? Please clarify it in the paper. Also, what’s the “Aj” in Table 2?

Author Response

(The authors gave the same response as above.)

Reviewer 3 Report

Need to add and discuss more references from the last three years.

The novelty statment should be given in more details.

The difference between the proposed paper and previous papers should be discussed.

reference to all should be given

The sensitivity analysis if applicable should be added to the paper.

The simulation study should be added and results should be compared with the existing studies.

The real example should be added and discussed with the existing studies

What are the limitations of the study?

What are the potential applications of the proposed study?

Neutrosophic statistics is the extension of classical statistics and is applied when the data is coming from a complex process or from an uncertain environment. The current study can be extended using neutrosophic statistics as future research. The statement that the proposed study can be extended for neutrosophic statistics can be added by citing some papers on neutrosophic statistics. 

Author Response

(The authors gave the same response as above.)

Round 2

Reviewer 3 Report

The paper can be accepted now